# Oxylipins Associated with D3-Creatine Muscle Mass/Weight and Physical Performance among Community-Dwelling Older Men

**DOI:** 10.3390/ijms232112857

**Published:** 2022-10-25

**Authors:** Megan M. Marron, Eric S. Orwoll, Peggy M. Cawthon, Nancy E. Lane, Anne B. Newman, Jane A. Cauley

**Affiliations:** 1Department of Epidemiology, School of Public Health, University of Pittsburgh, Pittsburgh, PA 15213, USA; 2Division of Endocrinology, Diabetes and Clinical Nutrition, School of Medicine, Oregon Health & Science University, Portland, OR 97239, USA; 3Research Institute, California Pacific Medical Center, San Francisco, CA 94107, USA; 4Department of Epidemiology and Biostatistics, University of California, San Francisco, CA 94158, USA; 5Department of Medicine, University of California, Davis, Sacramento, CA 95817, USA; 6Departments of Medicine and Clinical and Translational Science, University of Pittsburgh, Pittsburgh, PA 15213, USA

**Keywords:** oxygenated lipids, mobility, aging

## Abstract

Poor physical function is highly prevalent with aging, and strongly associated with D3-creatine muscle mass/weight. Using metabolomics, we previously identified several triglycerides consisting mostly of polyunsaturated fatty acids that were higher in older adults with good mobility. Here, we sought to further investigate polyunsaturated fatty-acid-related metabolites, i.e., oxylipins, and their associations with D3-creatine muscle mass/weight, gait speed, grip strength, and the Short Physical Performance Battery among 463 older men from the Osteoporotic Fractures in Men Study (MrOS). Oxylipins were measured in fasting serum using liquid chromatography–mass spectrometry. Muscle mass was estimated using D3-creatine dilution and adjusted for body size. We used linear regression to determine oxylipins associated with D3-creatine muscle mass/weight and physical performance, while adjusting for age, education, physical activity, Western dietary pattern, fish oil supplementation, and multiple comparisons. Among 42 oxylipins, none were associated with grip strength and 3 were associated with the Short Physical Performance Battery. In contrast, 18 and 17 oxylipins were associated with D3-creatine muscle mass/weight and gait speed, respectively. A subset of associations between oxylipins and gait speed were partially attenuated by D3-creatine muscle mass/weight. Higher levels of fatty acid alcohol and ketone oxylipins tended to be most strongly associated with gait speed and D3-creatine muscle mass/weight, potentially reflecting anti-inflammatory activity from these select oxylipins in MrOS older men.

## 1. Introduction

Poor physical functioning, such as a slow gait, is highly prevalent among older adults [1] and predicts multiple major health outcomes, including incident disability [2,3,4,5], falls [6,7], hospitalization [8,9], and all-cause mortality [1,2,5,7,10,11]. Muscle mass, measured by the D3-creatine dilution method and divided by weight, has shown to be a strong predictor of physical functioning and incident mobility limitations and disability [12,13]. A better metabolic characterization of muscle mass and physical functioning may shed light on potential biologic pathways that influence mobility among older adults.

Using plasma metabolomics, we previously found 23 triglycerides associated with high versus low walking ability among community-dwelling older men and women [14]. The triglycerides that were significantly higher among older adults with better mobility consisted mostly of polyunsaturated fatty acids. Here, we continued this investigation of polyunsaturated fatty-acid-related metabolites and physical functioning by examining oxylipins, i.e., metabolites derived from the oxidation of polyunsaturated fatty acids, e.g., omega-3 or omega-6 fatty acids. Oxylipins are major mediators in the positive and negative effects of polyunsaturated fatty acids. For example, oxylipins have a major role in pathways of inflammation and oxidative stress [15]. These oxygenated lipids may indicate metabolic pathways becoming altered with aging and morbidity that influence mobility.

In this report, we examined 42 oxylipins measured in fasting serum among 463 community-dwelling older men from the Osteoporotic Fractures in Men Study (MrOS). Using this information, we sought to (1) identify oxylipins associated with D3-creatine muscle mass/weight and/or physical performance (gait speed, grip strength, and the Short Physical Performance Battery) and (2) determine the extent to which D3-creatine muscle mass/weight attenuates associations between oxylipins and physical performance. We hypothesized that higher levels of more anti-inflammatory oxylipins and lower levels of more proinflammatory oxylipins would be associated with higher D3-creatine muscle mass/weight and better physical performance among this cohort of older men.

## 2. Results

Participants were 84 years old, on average, and the vast majority were white (90%) and had more than a high school education (81%). Those in the fastest gait speed tertile were younger, more educated, more physically active, taking fewer medications, and less likely to have heart disease and a history of stroke (Table 1). Men with faster gait speeds also had a lower body mass index, higher D3-creatine muscle mass/weight, and a better grip strength and score on the Short Physical Performance Battery (Table 1). Those in the fastest gait speed tertile tended to be less likely to have a Western dietary pattern, though the difference was marginally insignificant (*p* = 0.07). When examining dietary fat, there was no difference in percent kilocalories from fat, total fat, polyunsaturated fat, omega-6 fatty acids, nor omega-3 fatty acids by gait speed tertiles (*p*-values ranged from 0.60 to 0.998). There was also no difference in statin use by gait speed tertiles (*p* = 0.70). There was a smaller percentage of men taking fish oil supplementation among those in the slowest gait speed tertile when compared with the other tertiles, though this result was not statistically significant (*p* = 0.18).

Among the 42 oxylipins examined, 18 oxylipins were associated with D3-creatine muscle mass/weight and 17 oxylipins were associated with gait speed (FDR ≤ 10%). In contrast, none of the oxylipins were associated with grip strength and only three oxylipins were significantly associated with the Short Physical Performance Battery (Appendix A; false discovery rate (FDR) ≤ 10%). In sensitivity analyses, we found associations between oxylipins and the four outcomes were almost identical after additionally adjusting for non-steroidal anti-inflammatory medication use and statin use.

Almost all the oxylipins that were derived from the oxylipin generation pathway involving lipoxygenase (LOX) enzymes and arachidonic acid, an omega-6 fatty acid, were positively associated with gait speed (Figure 1), as well as positively associated with D3-creatine muscle mass/weight; however, fewer of these associations with D3-creatine muscle mass/weight reached statistical significance. All three of the oxylipins that were negatively associated with gait speed were derived from the pathway involving cytochrome P450 enzymes and either arachidonic acid or linoleic acid (omega-6 fatty acids; Figure 1).

Figure 2 is a volcano plot of the associations between the 42 oxylipins with respect to D3-creatine muscle mass/weight and physical performance, with a reference line at a *p*-value = 0.05. Almost all significant oxylipin associations with D3-creatine muscle mass/weight, gait speed, and the Short Physical Performance Battery were positive. Figure 3 is a heatmap illustrating the pattern of oxylipin associations across the four outcomes. Gait speed, the Short Physical Performance Battery, and D3-creatine muscle mass/weight appeared to be similar with regard to the pattern of their associations with oxylipins. These three outcomes tended to be positively associated with fatty acid alcohols, fatty acid ketones, and fatty acid epoxides, and they shared both positive and negative associations with fatty acid diols (Figure 3); though not all these associations reached statistical significance.

The amount of attenuation in the association between an oxylipin and gait speed (Figure 4a) or the Short Physical Performance Battery (Figure 4b) after additionally adjusting for D3-creatine muscle mass/weight is illustrated by the *y*-axis on Figure 4. Similarly, the *x*-axis on Figure 4 illustrates the amount of attenuation in the association between D3-creatine muscle mass/weight and gait speed (Figure 4a) or the Short Physical Performance Battery (Figure 4b) after additionally adjusting for an oxylipin. Among the 17 oxylipins significantly associated with gait speed, 11 were also associated with D3-creatine muscle mass/weight (*p* < 0.05; Appendix A). Adjusting for D3-creatine muscle mass/weight attenuated the associations between the eleven oxylipins and gait speed by 22–40% (*y*-axis on Figure 4a; Appendix A). In contrast, in separate models that adjusted for one of the eleven oxylipins, we found that the presence of an oxylipin attenuated the association between D3-creatine muscle mass/weight and gait speed by ≤5% (*x*-axis on Figure 4a; Appendix A).

All three of the oxylipins significantly associated with the Short Physical Performance Battery were also associated with D3-creatine muscle mass/weight (*p* < 0.05; Appendix A). Adjusting for D3-creatine muscle mass/weight attenuated the associations between the three oxylipins and the Short Physical Performance Battery by 28–33% (*y*-axis on Figure 4b; Appendix A). In contrast, in separate models that adjusted for one of the three oxylipins, we found that the presence of an oxylipin attenuated the association between D3-creatine muscle mass/weight and gait speed by ≤4% (*x*-axis on Figure 4b; Appendix A).

## 3. Discussion

In this community-dwelling cohort of mostly white men aged 78–98, we had information on 42 oxylipins detected in fasting serum. When examining physical performance and D3-creatine muscle mass/weight, we found oxylipins were most strongly correlated with gait speed and D3-creatine muscle mass/weight, followed by the Short Physical Performance Battery. None of the oxylipins were associated with grip strength. D3-creatine muscle mass/weight partially attenuated associations between a subset of oxylipins and physical performance.

To our knowledge, no previous study has examined associations between oxylipins and physical performance and very few have examined oxylipins in relation to body composition. Among a cohort of 43 older adults (mean age: 81 ± 0.9), 11 oxylipins were associated with a decline in appendicular lean mass over seven years of follow-up in unadjusted models [16]. Among the eleven oxylipins identified in the previous report, four omega-6 fatty-acid-derived oxylipins (9-HETE, 11-HETE, 15-HETE, and 8S-HETE) and three omega-3 fatty-acid-derived oxylipins (9-HEPE, 12S-HEPE, and 14-HDoHE) were also identified in the current cross-sectional analysis, but they were instead associated with a faster gait speed and higher D3-creatine muscle mass/weight among the current report’s older men. This discrepancy in results may be due to differences in demographics and health status among participants in the two cohorts. For example, participants in the previous report were recruited from France, had neither cardiovascular disease, diabetes, nor obesity at baseline, and were slightly younger than the current report’s men living in the U.S. In addition, the previous report used dual-energy X-ray absorptiometry, which does not measure muscle mass directly, but rather measures lean mass, which includes muscle, fibrotic and connective tissue, and bodily fluids [12]. Dual-energy X-ray absorptiometry lean mass also does not predict multiple major health outcomes such as D3-creatine muscle mass/weight [13]. Additionally, the previous report did not adjust for potential confounders (e.g., age) of the associations between oxylipins and change in appendicular lean mass, and they measured oxylipins in plasma, whereas oxylipins were measured in serum in the MrOS men. Future studies with similar information on oxylipins, muscle mass, and physical performance and analytic methods will be needed to determine the replicability of these results.

We previously found several triglycerides, consisting mostly of polyunsaturated fatty acids, were higher among older adults with high versus low walking ability [14]. In this report, we further examined polyunsaturated fatty-acid-related metabolites, i.e., oxylipins, in relation to physical performance and D3-creatine muscle mass/weight. Here, we found several oxylipins of the fatty acid alcohol and ketone classes were positively associated with gait speed and D3-creatine muscle mass/weight. These oxylipins are derived from either an omega-6 fatty acid precursor (arachidonic acid or linoleic acid) or an omega-3 fatty acid precursor (eicosapentaenoic acid or docosahexaenoic acid) all via the oxylipin-formation pathway involving lipoxygenases [15]. Lipoxygenases are enzymes expressed in immune, epithelial, and tumor cells that are involved in inflammation and tumorigenesis, as well as multiple other physiologic functions [17]. Much less is known regarding the beneficial effects of lipoxygenases and its oxylipin derivatives, though it is thought that they also display anti-inflammatory functions [17]. Higher levels of lipoxygenases-derived oxylipins among the MrOS men with faster gait speeds and higher D3-creatine muscle mass/weight may suggest that more of these oxylipins’ anti-inflammatory properties are activated.

Half of the oxylipins associated with gait speed are derived from arachidonic acid, an omega-6 polyunsaturated fatty acid that comes from dietary sources such as poultry, fish, and eggs, or from endogenous conversion of linoleic acid, an essential omega-6 polyunsaturated fatty acid found in plant oils [18]. The arachidonic-acid-derived oxylipins were mostly hydroxyeicosatetraenoic acids (HETEs). HETEs are formed by lipoxygenase enzymes, have previously been found to be associated with cardiometabolic diseases, and display both anti- and proinflammatory activity [15]. While it has been suggested that HETEs are potentially mostly proinflammatory [15], we found higher levels of 5-HETE, 8S-HETE, 9-HETE, 11-HETE, and 15-HETE associated with faster gait speed among the MrOS older men in multivariable adjusted models. Similarly, we also found two keto-octadecadienoic acid (KODE) oxylipins, 9-KODE and 13-KODE, positively associated with gait speed in our cohort. Similar to HETEs, KODEs are also derived from lipoxygenase enzymes, but with an omega-6 fatty acid, linoleic acid, as its precursor. KODEs are metabolites of hydroxyoctadecadienoic acid (HODEs), which have mostly been linked to pathological conditions [15]. Similar to HETEs, both KODEs and HODEs may have anti-inflammatory properties; however, information regarding their beneficial effects is limited. Future studies are needed to further explore the anti-inflammatory roles of these oxylipins and determine their potential beneficial impact on aging-related mobility.

Only three oxylipins were negatively associated with gait speed: 9,10- and 12,13-dihydroxyoctadecenoic acid (9,10-DiHOME and 12,13-DiHOME, respectively) and 14,15-dihydroxyeicosa-5,8,11-trienoic acid (14,15-DiHETrE). These three oxylipins are fatty acid diols and are derived via the cytochrome P450 pathway with an omega-6 fatty acid precursor (arachidonic acid or linoleic acid). Similar to other oxylipins already discussed, 9(10)- and 12(13)- epoxy-octadecenoic acid (EpOME) are mediators of inflammation and have toxic effects once converted into 9,10-DiHOME and 12,13-DiHOME [15]. The 14,15-DiHETrE oxylipin also has proinflammatory activity [19,20]. Using untargeted metabolomics, 14,15-DiHETrE was found to be higher among patients with sarcoidosis, a multiorgan disease characterized by systemic inflammation, when compared with healthy controls [19]. Higher levels of 14,15-DiHETrE, as well as 9,10-DiHOME and 12,13-DiHOME, may reflect greater systemic inflammation among MrOS older men with worse mobility.

Since this was only a cross-sectional study, mediation and temporality could not be determined. With these limitations in mind, we informally examined statistical mediation models by determining whether adjusting for an oxylipin would attenuate a portion of the association between D3-creatine muscle mass/weight and physical performance; however, we found minimal attenuation (≤5%). On the other hand, when adjusting for D3-creatine muscle mass/weight, we saw attenuations in the association between a subset of oxylipins and physical performance ranging from 22% to 40%. D3-creatine muscle mass/weight may be attenuating a larger proportion of the association between oxylipins and physical performance (when compared to the amount of attenuation observed in the association between D3-creatine muscle mass/weight and physical performance after adjusting for an oxylipin) because D3-creatine muscle mass/weight is a more global measure when compared to a single oxylipin [14]. Future longitudinal studies are needed to determine temporality and mediation between multiple oxylipins simultaneously, muscle mass, and physical performance among older adults. It should also be noted that a subset of oxylipins were associated with physical performance independently of D3-creatine muscle mass/weight, of which most were fatty acid diols; this finding potentially suggests that other pathways may also exist that do not act through muscle mass to have an impact on mobility.

Oxylipins are bioactive lipids that are not stored but are derived de novo as needed from the oxidation of polyunsaturated fatty acids [21]. An oxylipin profile depends on the amount of dietary polyunsaturated fatty acids and enzymes present, and the affinity of the enzymes for the polyunsaturated fatty acids present [21]. An oxylipin profile can also be altered by certain pharmaceuticals, e.g., non-steroidal anti-inflammatory drugs, antithrombotic drugs, and zileuton (an asthma medication) [22]. Through autocrine and paracrine signaling, oxylipins are major mediators in both the positive and negative effects of polyunsaturated fatty acids [15]. For example, oxylipins are involved in immunity, oxidative stress, apoptosis, tissue repair, blood clotting, pain, cell proliferation, and vascular functioning, and have both proinflammatory and anti-inflammatory properties [15,21,23,24]. It is well-known that inflammation is associated with worse walking ability and physical disability [25,26,27]. In fact, increases in interleukin-6, a proinflammatory cytokine, was shown to mirror age-related decline in gait speed among community-dwelling older men and women [28]. Thus, unsurprisingly, multiple oxylipins were significantly associated with gait speed in this report. We know age-related decline in gait speed has etiologies that can be broadly classified as musculoskeletal, of course, but also as neurologic and cardiorespiratory. Our report found that oxylipins were also significantly associated with D3-creatine muscle mass/weight and previous studies have shown relationships between oxylipins and cardiovascular diseases [21] and Alzheimer’s disease and related dementia [29]. Thus, there are likely multiple mechanisms and physiological systems that oxylipins act through to impact gait speed with aging. Modulating the oxylipin profile has potential promise since it may have a global beneficial effect across multiple physiological systems, and as a result preserve mobility with aging.

None of the oxylipins examined in this report were significantly associated with grip strength, whereas 40% of the oxylipins were significantly associated with gait speed. Grip strength is influenced by the musculoskeletal system and thought of as a marker of sarcopenia [30], whereas in the previous paragraph we discussed how gait speed is a complex phenotype influenced by multiple physiological systems [30,31]. The influence of multiple physiological systems may indicate that lipids and lipid mediators have multiple pathways in which they could affect gait speed, potentially translating to larger effect sizes that are easier to detect when compared with associations with grip strength. Thus, a limitation of our study may have been that it was underpowered to detect smaller effect sizes between oxylipins and grip strength. In addition, we found multiple oxylipins associated with D3-creatine muscle mass/weight. D3-creatine muscle mass/weight is a more precise and direct marker of muscle mass when compared with grip strength, which may explain why we observed significant associations with D3-creatine muscle mass/weight, but not with grip strength. Other limitations of this work include the cross-sectional nature, restricting us from assessing temporality between oxylipins, D3-creatine muscle mass/weight, and physical performance. In addition, reference ranges indicating healthy concentrations of oxylipins are unknown; therefore, we were unable to determine whether low or high values for oxylipins in our cohort would be considered outside of a normal range. Despite these limitations, our report has several strengths: we had information on several oxylipins from a well-characterized cohort of community-dwelling older men. In addition, we had objective measurements of physical performance and muscle mass measured by the precise D3-creatine dilution method.

Several oxylipins were positively associated and few were negatively associated with physical performance and D3-creatine muscle mass/weight among the MrOS cohort of older men. A subset of oxylipins, mostly fatty acid diols, were associated with physical performance independently of D3-creatine muscle mass/weight. These results are a step toward a better understanding of the metabolic mechanisms contributing to muscle health and physical functioning with aging. Future research needs to determine whether these oxylipins also predict trajectories of muscle mass and gait speed with aging and how likely they are to be causal factors of differences among older adults with varying physical functioning capacities. Gaining knowledge on the metabolic mechanisms and causal chain of how oxylipins are related to muscle mass and physical performance among older adults may indicate pathways to intervene on to prevent or reduce progression of disability and promote healthy aging in the population.

## 4. Materials and Methods

### 4.1. The Osteoporotic Fractures in Men (MrOS) Study

The MrOS study is a prospective longitudinal cohort of 5994 community-dwelling men ages 65 and older [32,33]. From March 2000 to April 2002, participants were recruited from six study sites across the United States: Birmingham, AL; Minneapolis, MN; the Monongahela Valley near Pittsburgh, PA; Palo Alto, CA; Portland, OR; and San Diego, CA. Ineligibility criteria included a history of bilateral hip replacement or the inability to walk without assistance from another person. The study was originally designed to determine risk factors of incident fractures and the effects of fractures on quality of life among older men. The institutional review board at each participating university approved the MrOS study protocol. All participants provided written informed consent.

Analytic sample: During 2014–2016, 2786 MrOS men were contacted for a fourth visit (year 14). Among these surviving participants, 1841 returned for a clinic exam, 583 only completed questionnaires, and 362 refused participation (Appendix A). A microbiome ancillary study was undertaken at the fourth visit [34] and included the first 600 MrOS men who volunteered to participate. Among those 600 men, 568 also had at least five stored serum samples for oxylipin measurements. Oxylipin measures from 21 participants were excluded due to samples failing assay quality control. We also excluded eight participants who did not fast for at least eight hours prior to phlebotomy. Among the remaining 539 participants with fasting oxylipin data, we excluded 18 participants missing physical performance measures (gait speed, grip strength, and/or the Short Physical Performance Battery), 55 participants missing D3-creatine muscle mass/weight, and 3 participants missing D3-creatine muscle mass/weight and a physical performance measure. Thus, our analytic sample included 463 MrOS men with information on fasting oxylipins, D3-creatine muscle mass/weight, and physical performance (Appendix A). When compared with the remaining 1378 men who returned for a clinic exam during the fourth MrOS visit, the 463 men in this report were, on average, 0.6 years younger, more physically active, consumed about 5 g less in total dietary fat per week, and scored 0.44 points higher on the Short Physical Performance Battery (*p*-values < 0.05). There were no differences in race, education, smoking status, total number of medications, statin use, fish oil use, percentage of kilocalories from dietary fat, weight, body mass index, muscle mass, muscle mass/weight, grip strength, nor gait speed, nor the likelihood of heart disease, stroke, hypertension, diabetes, chronic obstructive pulmonary disease, nor osteoarthritis among the 463 men included in this report versus the 1378 other men who returned for a clinic exam at the fourth MrOS visit.

### 4.2. Oxylipins

The John Newman laboratory within the West Coast Metabolomics Center at the University of California, Davis, received serum samples from MrOS in December 2016 and measured oxylipins using liquid chromatography–mass spectrometry [35,36] with methods similar to those used in previous reports [37]. Oxylipins are potent and short-lived metabolites with many factors impacting them, including diet [21]. Thus, to minimize biases, we only included information on oxylipins that were measured using serum samples collected after a fast of at least eight hours. The oxylipin standard panel measures 76 oxylipins in serum (Appendix A). However, 34 oxylipins were excluded from analyses because they were detected in less than 80% of participants [38] (Appendix A). Among the remaining 42 oxylipins (Appendix A), 5 were detected in all 463 participants and 37 were detected in at least 80% of the participants, of which missing values were assumed to be due to the true values being below the detectable limit [39] and were replaced with half the minimum recorded value for the respective oxylipin [39]. Among the 42 oxylipins examined in this report, 16 were fatty acid alcohols, 13 were fatty acid diols, 8 were fatty acid epoxides, 3 were fatty acid ketones, 1 was a fatty acid triol, and 1 was a nitro-fatty acid. Figure 1 illustrates the enzyme pathway and polyunsaturated fatty acid precursor [15] of these oxylipins.

### 4.3. D3-Creatine Muscle Mass

D3-creatine muscle mass was estimated using the D3-creatine dilution method and adjusted for body size using weight measured in the clinic [12]. Participants ingested a 30 mg dose of stable isotope-labeled creatine (D3-creatine) and provided an overnight-fasting urine sample approximately 3–6 days later. Using liquid chromatography–mass spectroscopy, D3-creatine, unlabeled creatinine, and creatine were measured in the urine, which was used to estimate total body creatine pool size and skeletal muscle mass [40].

### 4.4. Physical Performance

Gait speed was the best speed from two trials, with participants walking their usual pace for six meters. Grip strength was the maximum of two trials of both hands using a JAMAR dynamometer (Sammons Preston Rolyan, Bolingbrook, IL, USA). Participants’ ability and time to stand up from a chair five times in a row without using their arms was assessed. Balance was assessed as ability to stand in a side-by-side stance, tandem stance, and semi-tandem stance. Using gait speed, the chair stand test, and the balance tests, we calculated the Short Physical Performance Battery [41], which ranges from 0 (lowest score) to 12 (highest score).

### 4.5. Participant Characteristics

Participants completed a health history questionnaire and self-reported their date of birth, which was used to calculate age and their race, education, smoking status, and history or presence of heart disease (myocardial infarction, angina, and/or congestive heart failure), stroke (or transient ischemic attack), hypertension, diabetes, chronic obstructive pulmonary disease, and/or osteoarthritis. Participants brought in all medications taken in the past 30 days to their clinic visit for an inventory. Physical activity was assessed from the self-report questionnaire, the Physical Activity Scale for the Elderly [42], where a higher score indicates higher physical activity. Body mass index was calculated from height and weight measured at the clinic visit. Dietary information was self-reported using a Food Frequency Questionnaire adapted by the Block Dietary Systems for the MrOS. The adapted Food Frequency Questionnaire was a brief version for older adults based on dietary recall data from the Third National Health and Nutrition Examination Survey. Using this information, a Western dietary pattern was previously derived using factor analysis [43]. We examined a Western dietary pattern that was high in red meat, fried foods, and high-fat dairy, as well as a healthy dietary pattern that was high in fruits, vegetables, whole grains, and lean meats [43].

### 4.6. Statistical Analysis

All 463 participants had complete data on D3-creatine muscle mass/weight and physical performance. Complete data on additional study characteristics were available on 402 (88%) participants. Missing values were replaced using multiple imputation and available information on age, education, chronic conditions, number of medications, physical activity, and systolic blood pressure. We replaced missing values with the average of five imputed values for only two variables of interest, dietary information (1% missing) and smoking status (10% missing).

Mean (standard deviation) or frequency (percent) was used to describe differences among the 463 men by tertiles of gait speed. Differences by gait speed tertiles were tested using analysis of variance for normally distributed continuous measures and chi-square tests for categorical measures. Pairwise comparisons were examined when overall differences were observed at a 0.05 significance level.

Log-transformed oxylipins and all other continuous variables were standardized to a mean of zero and standard deviation of one prior to analyses. Linear regression models were examined to determine oxylipins associated with the following outcomes: gait speed, grip strength, Short Physical Performance Battery, and D3-creatine muscle mass/weight. Each linear regression model contained a single oxylipin and a single outcome, while adjusting for age, more than a high school education, the Physical Activity Scale for the Elderly score, the Western style dietary pattern score, and fish oil supplementation use. We used a Benjamini–Hochberg correction [44] to account for multiple comparisons, with a 10% FDR. We plotted the multivariable adjusted associations between oxylipins and physical performance and D3-creatine muscle mass/weight using volcano plots, where the *y*-axis illustrates the significance in the association between an oxylipin and either a physical performance measure or D3-creatine muscle mass/weight and the *x*-axis illustrates the magnitude of the standardized beta coefficient for the respective association. Each dot on the volcano plot represents a single oxylipin, color-coded by its fatty acid classification. We also plotted the multivariable adjusted associations between oxylipins and physical performance and D3-creatine muscle mass/weight on a heatmap to illustrate the pattern of oxylipins associated with the four outcomes. The heatmap includes a dendrogram of D3-creatine muscle mass/weight and physical performance, computed in R using its default hierarchical clustering based on Euclidean distance.

We examined the extent of shared variance between oxylipin and D3-creatine muscle mass/weight with respect to gait speed by determining the percentage of attenuation in the multivariable adjusted association between an oxylipin and gait speed after additionally adjusting for D3-creatine muscle mass/weight. We used the following steps: (1) identified the subset of oxylipins associated with gait speed (*p* < 0.05; FDR < 10%); (2) determined the subset of gait-speed-associated oxylipins that were also associated with D3-creatine muscle mass/weight (*p* < 0.05); (3) determined whether D3-creatine muscle mass/weight was associated with gait speed (*p* < 0.05); and (4) calculated percentage of attenuation as 100 * [(b1−b2)/b1], where b1 was the coefficient from a linear regression model of gait speed on an oxylipin, adjusting for age, education, physical activity, Western dietary pattern, and fish oil use and b2 was the same coefficient after further adjusting for D3-creatine muscle mass/weight. We also examined the extent to which an oxylipin attenuated the association between D3-creatine muscle mass/weight and gait speed. We used a scatterplot to illustrate these percentage of attenuations, where the *y*-axis is the percentage of attenuation in the multivariable adjusted association between an oxylipin and gait speed when additionally adjusting for D3-creatine muscle mass/weight and the *x*-axis is the percentage of attenuation in the multivariable adjusted association between D3-creatine muscle mass/weight and gait speed when additionally adjusting for the respective oxylipin. Lastly, we repeated these steps for the Short Physical Performance Battery.

## Figures and Tables

**Figure 1 ijms-23-12857-f001:**
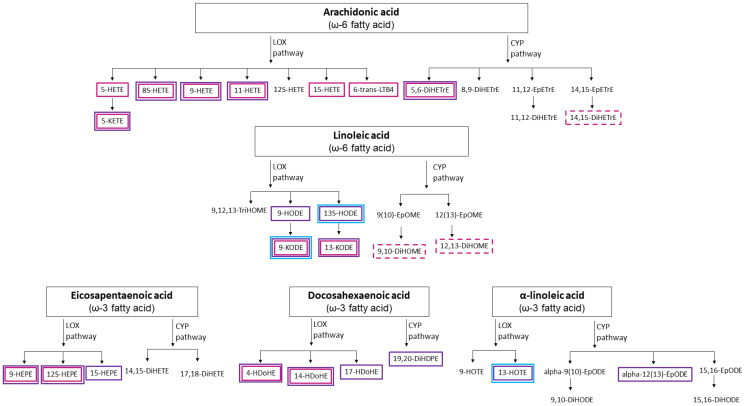
Polyunsaturated fatty acids and metabolic pathways that create the oxylipins examined in this report. Squares indicate significant associations (*p* < 0.05, FDR ≤ 10%; Appendix A) between oxylipins and D3-creatine muscle mass/weight (purple squares), gait speed (pink squares), or the Short Physical Performance Battery (blue squares). Non-dotted squares indicate positive associations and dotted squares indicate negative associations. Abbreviations: LOX—lipoxygenase; CYP—cytochrome P450 (oxylipin abbreviations are listed in Appendix A). Not pictured: 10-Nitrooleate; 9,10-e-DiHO; and 9,10-EpO.

**Figure 2 ijms-23-12857-f002:**
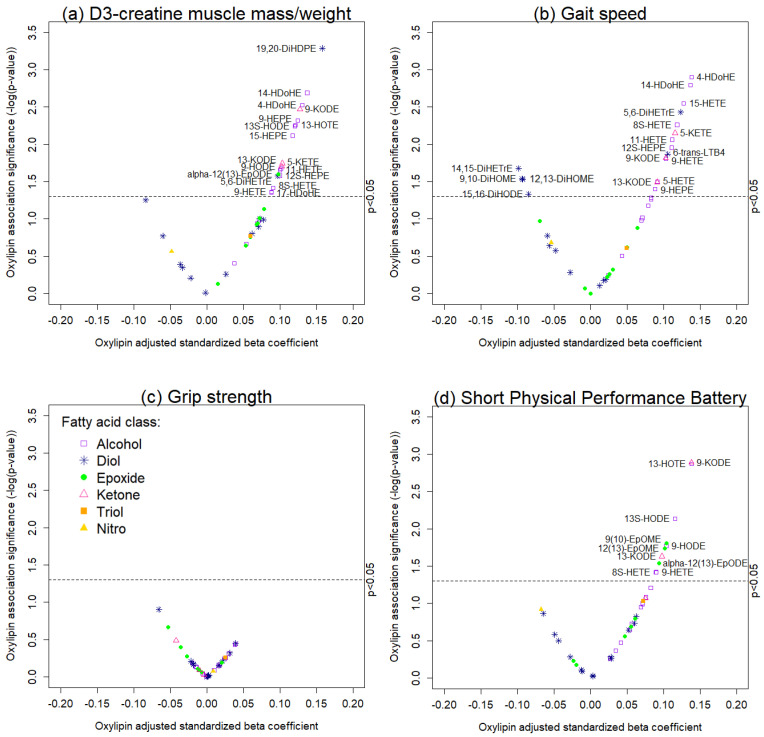
Significance vs. multivariable adjusted association between oxylipins and (**a**) D3-creatine muscle mass/weight, (**b**) gait speed, (**c**) grip strength, and (**d**) the Short Physical Performance Battery among 463 MrOS men. Models were adjusted for age, more than high school education, Physical Activity Scale for the Elderly, Western style dietary pattern score, and fish oil supplement use. Oxylipin abbreviations are listed in Appendix A.

**Figure 3 ijms-23-12857-f003:**
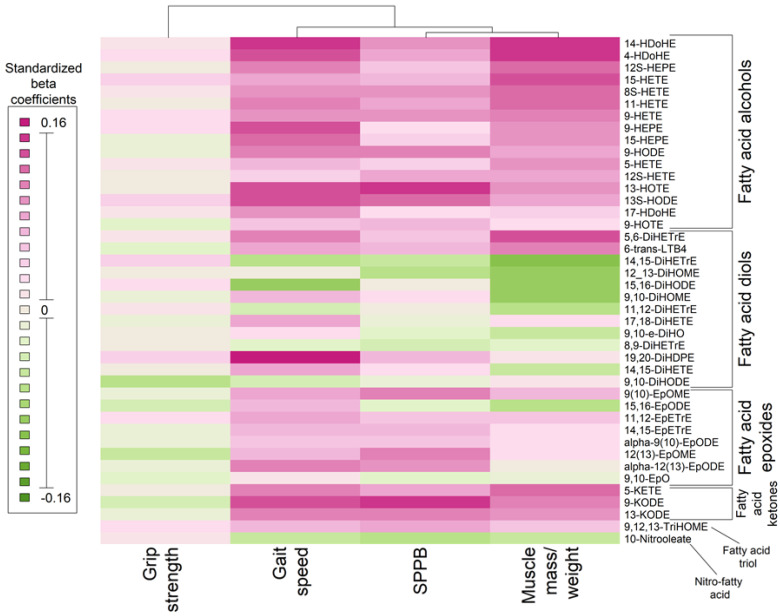
Heatmap of multivariable adjusted associations between oxylipins and physical performance and D3-creatine muscle mass/weight and a dendrogram of hierarchical clustering of the four outcomes among 463 MrOS men. Models were adjusted for age, more than high school education, Physical Activity Scale for the Elderly, Western style dietary pattern score, and fish oil supplementation. Oxylipin abbreviations are listed in Appendix A.

**Figure 4 ijms-23-12857-f004:**
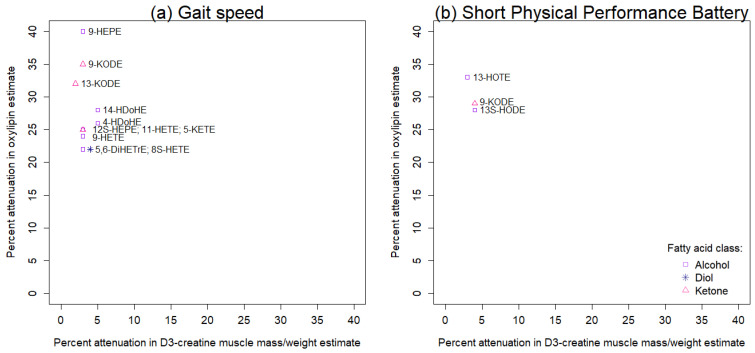
Attenuation in multivariable adjusted associations between select oxylipins and (**a**) gait speed or (**b**) the Short Physical Performance Battery after additionally adjusting for D3-creatine muscle mass/weight versus attenuation in multivariable adjusted associations between D3-creatine muscle mass/weight and (**a**) gait speed or (**b**) the Short Physical Performance Battery after additionally adjusting for an oxylipin among 463 MrOS men. Models were initially adjusted for age, more than high school education, Physical Activity Scale for the Elderly, Western style dietary pattern score, and fish oil supplementation. Oxylipin abbreviations are listed in Appendix A.

**Table 1 ijms-23-12857-t001:** Characteristics among N = 463 MrOS men by tertiles of gait speed.

Mean (SD) or Frequency (%)	Gait Speed Tertiles			*p*-Values ^1^
	Slowest (T1) *n* = 141	Average (T2) *n* = 155	Fastest (T3) *n* = 167	
Gait speed (m/s)	0.8 (0.1) Range: 0.40, 0.99	1.1 (0.06) Range: 1.0, 1.2	1.4 (0.1) Range: 1.21, 1.88	----
Age	85 (4)	84 (4)	83 (3)	<0.0001; T1 > T2 > T3
White race	129 (91%)	139 (90%)	148 (89%)	0.71
More than high school education/equivalency	104 (74%)	126 (81%)	147 (88%)	0.006; T1 < T3
Never smoked cigarettes	53 (38%)	62 (40%)	81 (49%)	0.12
Physical Activity Scale for the Elderly	99 (63)	130 (68)	145 (62)	<0.0001; T1 < T2 < T3
Chronic conditions:				
Heart disease (MI, angina, CHF)	34 (24%)	38 (25%)	21 (13%)	0.01; T1,T2 > T3
Stroke (or TIA)	25 (18%)	18 (12%)	14 (8%)	0.04; T1 > T3
Hypertension	75 (54%)	84 (54%)	79 (47%)	0.38
Diabetes	27 (19%)	25 (16%)	17 (10%)	0.08
COPD	19 (13%)	17 (11%)	15 (9%)	0.45
Osteoarthritis	41 (29%)	39 (25%)	33 (20%)	0.16
Total number of medications	9.4 (5)	8.9 (5)	8.2 (4)	0.04; T1 > T3
Taking statins	80 (57%)	83 (54%)	97 (58%)	0.70
Taking non-steroidal anti-inflammatory medications	17 (12%)	29 (19%)	22 (13%)	0.21
Taking fish oil	22 (16%)	37 (24%)	37 (22%)	0.18
Dietary information per day:				
Western dietary pattern ^2^	0.005 (1)	−0.07 (0.9)	−0.24 (1)	0.07
% Kcal from fat	40 (7)	41 (7)	41 (7)	0.68
Total fat (g)	68 (31)	68 (33)	68 (32)	0.998
Polyunsaturated fat (g)	16 (8)	17 (9)	17 (9)	0.60
Height (cm)	172 (7)	172 (7)	173 (6)	0.44
Weight (kg)	83 (14)	79 (13)	79 (13)	0.005; T1 > T2,T3
Body mass index (kg/m^2^)	28 (4)	27 (4)	26 (4)	0.0002; T1 > T2,T3
D3-creatine muscle mass (kg)	23 (4)	24 (4)	25 (4)	<0.0001; T1,T2 < T3
D3-creatine muscle mass/weight (%)	28 (4)	31 (5)	32 (5)	<0.0001; T1 < T2 < T3
Grip strength (kg)	33 (7)	36 (8)	39 (7)	<0.0001; T1 < T2 < T3
Short Physical Performance Battery	7.6 (3)	10 (2)	11 (2)	<0.0001; T1 < T2 < T3

^1^ When *p* < 0.05, pairwise comparisons were made to determine which tertiles differed; ^2^ dietary pattern high in red meat, fried foods, and high fat dairy.

## Data Availability

Data can be requested at the MrOS Online website: https://mrosonline.ucsf.edu/ (accessed on 20 March 2019).

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
