# Peer review of "Oxylipins Associated with D3-Creatine Muscle Mass/Weight and Physical Performance among Community-Dwelling Older Men"

_ijms, 2022, doi:10.3390/ijms232112857_

Round 1

Reviewer 1 Report

The authors present an interesting paper looking at metabolites associated with muscle mass and gait speed in older men. The authors have done previous work in metabolites and gait speed, and this exploratory study is a follow on from that, looking at oxylipins derived from PUFA oxidation. Thus, these are largely unstudied and there appears to be very limited existing data in this area.

This is a novel exploratory piece of work, with some clear figures which are well explained, and understandable. The paper lacks discussion on the mechanisms of how the metabolites associate with muscle mass and gait speed, and possible direction of such associations – if these are all unknown, it would be worth including that information.

The authors hypothesise that the metabolites that are associated with muscle mass and gait speed will be ones which are involved in inflammation/oxidative stress, and that physical performance will be associated with lower pro-inflammatory and higher anti-inflammatory oxylipins. This does not turn out to be the case for all tested oxylipins, and there is limited understanding of why some results are unexpectedly the other way around.

Does the timing of the test have an influence? Could authors include a comment on how stable these metabolites are? How reproducible are samples ? Do we know how much recent diet influence the results – or are they usually stable over long periods of time? Do they vary depending on time of sampling? Etc.

Sime of the findings do not agree with existing evidence – for example oxylipins associated with decline in appendicular lean mass in existing paper, versus a higher D3 creatinine mass/weight in this study. The authors do not shy away from this, but I wondered whether all possible explanations for this have been explored?

I wondered why none of the oxylipins were associated with grip strength, which seems a bit odd. Do the authors have any idea why this might be the case? Could they include a comment on this ?

The discussion lacks a description of the impact of these results, what are the next steps? What is the potential impact of gaining further understanding of these metabolites ?

Some specific points

Line 73 – I would suggest describing non-significant associations as a trend.

Line 77-79 – I am not sure such a conclusion can be drawn with p=0.18
Line 82 – write FDR out in full for first use

Line 172-173 – I think many researchers believe that DEXA does measure muscle mass, so further explanation is warranted here to explain this statement.

Line 209-210 – I think a reference should be included here.

Line 212-214 – consider rephrasing , seems contradictory

Line 252 – 254 is repetitive, there are a few bits of repetition throughout paper.

Line 256-260 - I would question whether having data/samples available is a study strength, consider revising

Line 367-368 could be more concise

Reviewer 2 Report

The authors made in an early stage the remark that they identified previously higher plasma levels triglycerides (TGs) levels, which consisted mostly of polyunsaturated fatty acids (PUFAs), in older adults with better mobility than in the corresponding age, but less mobile, subjects. However, this observational investigation was not able to link any of the lower limb muscle biochemical, physiological, or physical performance attributes to the systemic circulation of PUFAs or their metabolites. Also, it is worth noting that better-trained subjects present better muscle mass and lower systemic circulating triglycerides levels. Therefore, the claim of higher plasma TGs in better-trained older subjects might have been at odds with the evidence reported in the literature. And it is fair to say that the levels of plasma circulating fatty PUFAs are exclusively generated by the acquired diet. Since PUFAs cannot be synthesised in the body there should be no obvious association with muscle mass.

Table 1 clearly shows that unsurprisingly, the best physical condition was displayed by the subjects who had the highest muscle mass and the lowest prevalence of comorbidities in every single aspect.

Of note, the were no significant differences in PUFAs dietary provisions in the form of fish oil and polyunsaturated fat, and thereby that of oxylipins availability, across all groups.

On this background, the authors presently attempted to correlate/associate levels of a subset (42) of the intermediary fat metabolism (oxylipins) to muscle mass and markers of muscle strength and endurance. None of the oxylipins was associated with grip strength (a marker of muscle mass). However, and in line with the above comment, low muscle mass has often been associated with mobility concerns, because with less muscle comes a reduced function. This fact has been acknowledged by the authors too in the manuscript. Overall, the authors’ hypothesis of the exitance of an association between muscle mass and oxylipins could not be upheld.

Although all statins are quite effective in decreasing triglyceride levels and other free fatty acids, they do also increase the risk of developing muscle insulin resistance and muscle atrophy. Since almost 60% of the older men enrolled in this study were on statins, this medication would have been a strong confounder in determining both biochemical and functional muscle outcomes.

There is no account for the relevance of shifting the field of investigation from polyunsaturated fatty acids (previous study) to oxylipins (present study). The assertion that “oxylipins are major mediators in the positive and negative effects of polyunsaturated fatty acids” may be too generic/ambiguous as the former are overtly generated during inflammation and infection, thereby linking lipid metabolism with immunity and vascular functions. Unsurprisingly, oxylipins have been assigned important roles in the aetiology of cardiovascular dysfunctions or CVDs, including hypertension. And yet, the authors attempt presently to associate levels of circulating oxylipins to muscle mass and muscle function tests.

There might be some associations between them, but they were most likely not causative.

Overall, the authors ought to consider the above concerns as limitations of the study, and is, therefore, worth discussing in the Discussion section.

Round 2

Reviewer 2 Report

I am happy to recommend the manuscript be accepted in its present form.